# Association of Autism Onset, Epilepsy, and Behavior in a Community of Adults with Autism and Severe Intellectual Disability

**DOI:** 10.3390/brainsci10080486

**Published:** 2020-07-27

**Authors:** Stefano Damiani, Pietro Leali, Guido Nosari, Monica Caviglia, Mariangela V. Puci, Maria Cristina Monti, Natascia Brondino, Pierluigi Politi

**Affiliations:** 1Department of Brain and Behavioral Sciences, University of Pavia, 27100 Pavia, Italy; natascia.brondino@unipv.it (N.B.); pierluigi.politi@unipv.it (P.P.); 2Faculty of Medicine, University of Pavia, 27100 Pavia, Italy; leali.pietro@gmail.com; 3Department of Neurosciences and Mental Health, IRCCS Ca’ Granda Ospedale Maggiore Policlinico, 20122 Milan, Italy; guido.nosari@gmail.com; 4RSD Cascina Rossago, 27050 Ponte Nizza, Italy; mo.caviglia@gmail.com; 5Department of Public Health, Experimental and Forensic Medicine, University of Pavia, 27100 Pavia, Italy; mariangela.puci@unipv.it (M.V.P.); cristina.monti@unipv.it (M.C.M.)

**Keywords:** autism in adulthood, intellectual disability, regressive autism, epilepsy, challenging behaviors

## Abstract

Autism spectrum disorders (ASDs) are hard to characterize due to their clinical heterogeneity. Whether epilepsy and other highly prevalent comorbidities may be related to specific subphenotypes such as regressive ASD (i.e., the onset of symptoms after a period of apparently typical development) is controversial and yet to be determined. Such discrepancies may be related to the fact that age, level of cognitive functioning, and environmental variables are often not taken into account. We considered a sample of 20 subjects (i) between 20 and 55 years of age, (ii) with severe/profound intellectual disability, (iii) living in the same rural context of a farm community. As a primary aim, we tested for the association between epilepsy and regressive ASD. Secondly, we explored differences in behavioral and pharmacological profiles related to the presence of each of these conditions, as worse behavioral profiles have been separately associated with both epilepsy and regressive ASD in previous studies. An initial trend was observed for associations between the presence of epilepsy and regressive ASD (odds ratio: 5.33; 95% CI: 0.62–45.41, *p*-value: 0.086). Secondly, subjects with either regressive ASD or epilepsy showed worse behavioral profiles (despite the higher pharmacotherapy they received). These preliminary results, which need to be further confirmed, suggest the presence of specific associations of different clinical conditions in subjects with rarely investigated phenotypes.

## 1. Introduction

Autism spectrum disorder (ASD) is a neurodevelopmental condition whose phenotype encompasses various degrees of disability. Its growing prevalence has recently reached an estimate of 1 over 40 children [1]. ASD is associated with a wide range of comorbidities, among which intellectual disability (ID) or “low-functioning” ASD, epilepsy, and psychiatric disorders are the most frequent [2,3]. Compared to their “high-functioning” counterparts, low-functioning ASD individuals tend to exhibit more severe symptomatology and represent approximately 50% of the ASD population [4]. Few studies have been conducted on this 50% of the ASD population with comorbid ID, with its poor clinical outcomes and, thus, higher need for effective therapeutical strategies [5]. Similar to ID, epilepsy is also associated with challenging behaviors in individuals with ASD and ID [6]. Moreover, congenital autism spectrum disorders (CASD, when the core symptoms are evident from the very early life stages) may show different clinical presentations from regressive autism disorders (RASD, when the child loses abilities and milestones that were already achieved) [7]. In spite of the well-established presence of shared mechanisms between epilepsy and the autism spectrum being taken as a whole [8], the degree of association between epilepsy and RASD is not fully understood, with the few studies on this topic mostly focused on children [9,10].

A categorization of ASD subphenotypes is highly needed. This explorative study describes the clinical profile of 20 adults with ASD and severe/profound ID by assessing the presence of epilepsy and their history of RASD. The primary aim is to assess (i) the chance of these two conditions occurring together and (ii) the impact of each condition over the clinical picture in order to test whether epilepsy or RASD may be associated with the individuals’ behavior. Both these features are hence compared to individual behaviors and pharmacotherapy, expecting worse profiles in subjects with epilepsy or RASD.

## 2. Materials and Methods

### 2.1. Study Design

This work is an observational retrospective study. We conducted a thorough data collection for each subject in two ways: (i) The number of seizures and sleep-disturbed nights were collected from the clinical records of each subject from January 2012 to December 2017 and sorted in 72 monthly timepoints for a total of 6 years; (ii) at the end of this time-window, behavioral profiles were assessed with the aberrant behavior checklist (ABC) [11] and data concerning antipsychotic, antidepressant, benzodiazepine, and anticonvulsant therapy were recorded.

### 2.2. Participants

Participants were recruited from Cascina Rossago, a farm community located in a rural area in the province of Pavia, Italy. For each subject, the legal representative gave his informed consent for inclusion before participation in the study. The study was conducted in accordance with the Declaration of Helsinki, and the protocol was approved by the Ethics Committee of IRCSS San Matteo, Pavia, Italy. This facility setting is meant to promote the active involvement of the user in farming activities such as fruit harvesting, breeding, and gardening. Moreover, sports and artistic workshops such as painting and drawing, music courses, pottery, basketball, and pool swimming are provided. The location is surrounded by nature: a rural environment was chosen both to avoid external disturbing factors such as city-life distress and to favor natural circadian and seasonal rhythms. This community was considered ideal for the purpose of our research for multiple reasons. Firstly, it is committed to offering lifelong care and working plans to adult ASD patients with severe to profound ID. The same living environment, activities, and diet are hence available for all the subjects. Secondly, each patient was constantly monitored by healthcare professionals, with accurate and detailed medical records. The main sample characteristics are shown in Table 1. RASD and epilepsy were considered as primary factors to present more specific information regarding the subsamples.

### 2.3. Materials

*Epilepsy—*A subject was considered epileptic if the diagnosis from a specialized neurological center was present. The patient’s legal representatives were contacted to further confirm the anamnestic data and to provide the official documentation confirming the clinical history. The retrospective analysis of the data allowed us to reconstruct the number of seizures suffered by each subject monthly. Taking into account the numerical limits of the sample in contrast with the detailed temporal data, a graphic illustration of the epilepsy trends was produced as a useful tool to promptly understand and visualize the differences between the two cohorts.

*CASD/RASD—*In relation to the diagnosis of RASD, early life medical records were available for all the subjects. According to the commonly agreed definition [12], we classified patients who initially reached cognitive and behavioral milestones but experienced a setback before the age of 3 as RASD.

### 2.4. Pharmacological Treatments

In our sample, 18 of the 20 participants were on psychopharmacological therapy. Antipsychotics were necessary for 10 subjects to support educational interventions for challenging behaviors. Anticonvulsants and benzodiazepines were used to control seizures in the majority of cases. In subjects without epilepsy (4 cases), these drugs were administered with the aim of controlling impulsive and self-harming behaviors [13].

### 2.5. Data Analyses

To describe the sample, we calculated summary statistics that are expressed as means, standard deviations (SD), median and 25th–75th percentiles or percentages, as appropriate. A Shapiro–Wilk test was used to test the normality of the data. We used Fisher’s exact test for categorical variables and Student’s *t*-test or the Mann–Whitney test, as appropriate, for the quantitative ones. Odds ratio (OR) with 95% confidence interval (CI) was calculated to evaluate the associations between epilepsy and RASD/CASD. Statistical analysis was conducted using STATA/SE for Windows, version 15 (StataCorp, College Station, TX, USA). A *p*-value <0.05 was considered significant.

## 3. Results

The mean age of the whole sample was 37.5 ± 9.48 years (range 23–55 years), 15 (75%) were males. For all patients, the mean BMI was 25.30 ± 4.21 kg/m^2^. Information about drug therapy revealed that 60% of patients consumed antiepileptic drugs, 55% benzodiazepines, 30% antidepressants, and 50% psychiatric drugs (Table 1). Table 1 shows patient characteristics for all subjects, stratified by the presence of epilepsy and CASD/RASD conditions. The history of registered epileptic seizures is graphically reported in Figure 1.

Six subjects had no history of epilepsy or RASD, three had only epilepsy, three had only RASD, and eight had both epilepsy and RASD (Figure 2A). The odds ratio was computed to check for associations between the presence of epilepsy and RASD (OR: 5.33; 95% CI: 0.62–45.41, *p*-value: 0.086).

The ABC median subscores were higher in epileptic patients versus nonepileptic patients and in the RASD group versus the CASD group (Table 2). No statistically significant difference was appreciated, but a trend was observed, for which each condition tends to increase the severity of challenging behaviors (all the ABC median subscores are higher in both epilepsy and RASD groups; see Figure 2).

## 4. Discussion

We analyzed a sample of adult patients with ASD and ID who are stably living within the setting of a farm community. The main purpose of our study is to assess the association between autism onset and epilepsy and the impact of these conditions on behavioral profiles. Even though limitations such as the sample size did not allow the emergence of statistically significant results, the RASD subjects were 5 times more likely to also have a diagnosis of epilepsy.

We retrospectively measured the average of monthly epileptic events, i.e., any type of seizure, to better describe the presence of epileptic events and their frequency. RASD patients experienced a greater number of events, with fewer and shorter epilepsy-free periods. The three CASD subjects with epilepsy seemed to respond better to pharmacotherapy, with better control of the seizures.

Concerning behavioral profiles, even if differences in the ABC domains did not reach statistical significance, both RASD and epilepsy groups scored higher in each of these subscores. This trend is important, especially considering that the behavioral impairments are observed despite the higher pharmacotherapy that the RASD and epilepsy groups are undergoing. In fact, these subjects are more likely to consume not only anticonvulsants/mood stabilizers and benzodiazepines (which would be intuitive due to the epileptic risk), but also antipsychotics and antidepressants, which were initiated to better control problem behaviors and dysregulated moods, respectively. Autism and epilepsy share several neurobiological pathways [14]. An increased association between RASD and epilepsy may be determined by severe brain damage (for instance, after encephalitis, which is a risk factor for both autism [15] and epilepsy [16]) occurring in the first years of life. For instance, in our sample, two subjects suffered acute infection-related, early age neuroinflammation, after which seizures and regression arose.

The main limitation of this study is the low sample size, which does not allow us to hypothesize more complex assumptions based on the statistical significance of the findings due to excessively wide confidence intervals. The restrictive inclusion criteria and the same living and care conditions allowed to exclude several confounding factors, such as the variability given by differences in cognitive profiles, environments, diets, and therapeutical/educational approaches. However, high intersubject variability in behavioral and pharmacological profiles was measured despite these highly specific criteria.

## 5. Conclusions

Our findings show that (i) RASD and epilepsy tend to present together in a subsample of subjects with ASD and ID, (ii) RASD is associated with poorer control of epileptic symptoms, and (iii) RASD and epilepsy tend to worsen the behavioral prognosis, requiring more psychopharmacological therapies to compensate the clinical problems. Even though previous studies concerning the association between RASD and epilepsy yielded controversial results, the apparently contrasting findings may be attributed to the often-neglected differences between different life-phases (children versus adults) and levels of cognitive functioning. Carefully considering age and the presence of ID as influencing factors may hence help to shed light on this relatively uncharted but promising field of investigation in order to refine clinical classifications and tailor individualized treatments. These results invite us to further explore the subject, with the aim of determining whether the occurrence of different symptoms in autistic subpopulations may eventually lead to the individuation of ASD phenotypes.

## Figures and Tables

**Figure 1 brainsci-10-00486-f001:**
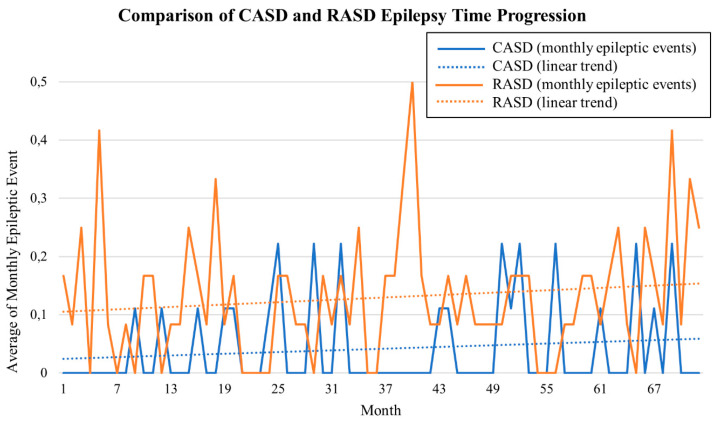
The graph represents oscillations in the presentation of epileptic events in CASD and RASD groups during the 6-year interval. The more frequent and higher spikes suggest a greater occurrence of seizures in the RASD group. Conversely, the seizure-free period reaches the zero level more often and for longer timespans in the CASD group. This trend tends to remain stable across the period of observation.

**Figure 2 brainsci-10-00486-f002:**
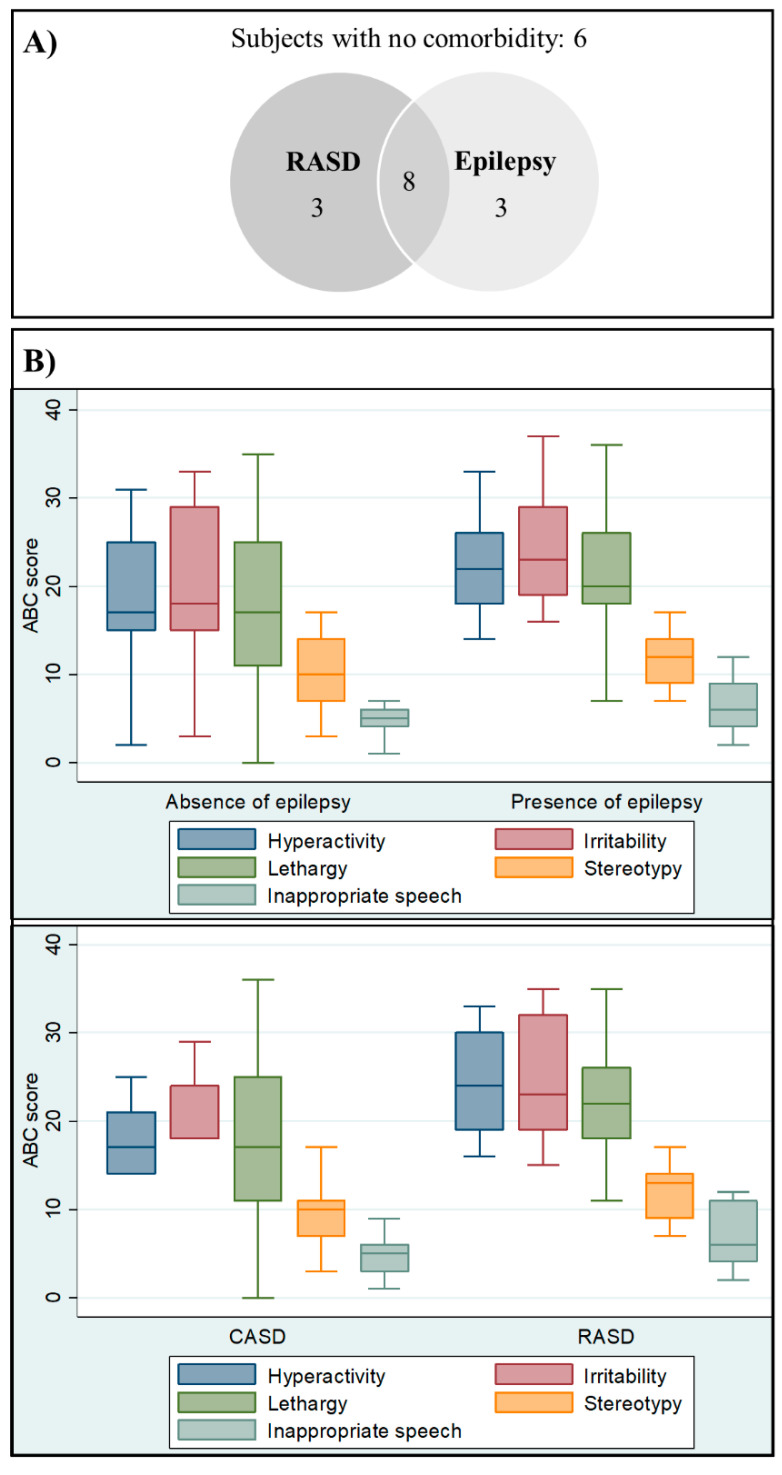
(**A**) Number of subjects with RASD, epilepsy, and both conditions are displayed. (**B**) Boxplots representing the difference of ABC median scores and the 25th–75th percentile of ABC scores in subjects stratified by epilepsy absent/present and CADS/RASD.

**Table 1 brainsci-10-00486-t001:** Sample characteristics.

	All Subjects	Groups Stratified by Absence/Presence of Epilepsy			Groups Stratified by Autism Onset		
	(n = 20)	Absence (n = 9)	Presence (n = 11)	Absence vs. Presence *p*-Value	CASD (n = 9)	RASD (n = 11)	CASD vs. RASD *p*-Value
Age—mean ± SD	37.5 ± 9.5	36.33 ± 3.9	38.5 ± 2.4	0.632 †	36. 6 ± 8.1	38.3 ± 10.8	0.698 †
Sex—n (%)							
*Male*	15 (75.0)	6 (66.7)	9 (81.8)	0.617 ^#^	5 (55.6)	10 (90.9)	0.127 ^#^
BMI (kg/m^2^)—mean ± SD	25.3 ± 4.2	24.8 ± 3.8	25.7 ± 4.7	0.649 †	26.1 ± 3.3	25.4 ± 4.6	0.553 †
Drug therapy—n (%)							
*anticonvulsant*	12 (60.0)	4 (44.4)	8 (72.7)	0.362 ^#^	4 (44.4)	8 (72.7)	0.362 ^#^
*benzodiazepine*	11 (55.0)	3 (33.3)	8 (72.7)	0.175 ^#^	3 (33.3)	8 (72.7)	0.175 ^#^
*Antidepressant*	6 (30.0)	2 (22.2)	4 (36.4)	0.642 ^#^	2 (22.2)	4 (36.4)	0.642 ^#^
*antipsychotic*	10 (50.0)	4 (44.4)	6 (54.6)	0.999 ^#^	4 (44.4)	6 (54.6)	0.999 ^#^

† Student’s *t*-test ^#^ Fisher’s exact test; CASD = congenital autism spectrum disorder; RASD = regressive autism spectrum disorder.

**Table 2 brainsci-10-00486-t002:** Aberrant behavior checklist (ABC) subscales scores.

	All Subjects	Groups Stratified by Absence/Presence of Epilepsy			Groups Stratified by Autism Onset		
	(n = 20)	Absence (n = 9)	Presence (n = 11)	Absence vs. Presence *p*-Value	CASD (n = 9)	RASD (n = 11)	CASD vs. RASD *p*-Value
Hyperactivity							
Mean ± SD	21.05 ± 7.56	19.0 ± 9.0	22.73 ± 6.08	0.270	17.56 ± 8.29	23.9 ± 5.80	0.056
Median (IQR)	21.0 (16.0−25.5)	17.0 (15.0−25.0)	22.0 (18.0−26.0)		17.0 (14.0−23.0)	24.0 (19.0−30.0)	
Irritability							
Mean ± SD	22.25 ± 8.65	19.67 ± 10.37	24.36 ± 6.73	0.209	19.56 ± 10.19	24.45 ± 6.87	0.261
Median (IQR)	21.5 (18.0−29.0)	18.0 (15.0−29.0)	23.0 (19.0−29.0)		18.0 (13.0−26.5)	23.0 (19.0−32.0)	
Lethargy							
Mean ± SD	20.65 ± 9.45	19.22 ± 11.34	21.82 ± 7.96	0.469	18.56 ± 11.86	22.36 ± 7.06	0.370
Median (IQR)	19.5 (16.5−26.0)	17.0 (11.0−25.0)	20.0 (18.0−26.0)		17.0 (9.0−29.5)	22.0 (18.0−26.0)	
Stereotypy							
Mean ± SD	11.15 ± 3.99	10.22 ± 4.71	11.91 ± 3.33	0.377	9.89 ± 4.51	12.18 ± 3.37	0.230
Median (IQR)	11.0 (7.0−14.0)	10.0 (7.0−14.0)	12.0 (9.0−14.0)		10.0 (7.0−13.0)	13.0 (9.0−14.0)	
Inappropriate speech							
Mean ± SD	5.85 ± 3.10	5.0 ± 2.92	6.55 ± 3.21	0.267	4.67 ± 2.50	6.82 ± 3.31	0.175
Median (IQR)	5.0 (4.0−7.5)	5.0 (4.0−6.0)	6.0 (4.0−9.0)		5 (2.5−6.5)	6.0 (4.0−11.0)	

*p*-values obtained from Mann–Whitney U-tests. CASD = congenital autism spectrum disorder; RASD = regressive autism spectrum disorder.

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
