# Peer review of "Association of Autism Onset, Epilepsy, and Behavior in a Community of Adults with Autism and Severe Intellectual Disability"

_brainsci, 2020, doi:10.3390/brainsci10080486_

Round 1

Reviewer 1 Report

The abstract does not even report the number of patients, very confused and does not convery the idea of what the autors want to comunicate, everything should be rewritten in detail.

Insufficient bibliography. Even the discussion of the results is insufficient, the conclusions do not exist, the authors themselves say that their results are preliminary and must confirmed by further study....

The tables are also completely rewritten.

Reviewer 2 Report

Thanks for your contribution to the topic. However , I am afraid that the entire manuscript needs to go through a thorough revision and editing of the english language. Unfortunately current presentation makes it very difficult to have a clear understanding of background, aims, and conclusions. Once the revision is completed, hopefully  the essential feature of the manuscript will become clearer. 

Author Response

Thank you, the manuscript has been thoroughly reworked according to the reviewer suggestions

Reviewer 3 Report

Association of autism onset, epilepsy and behavior in a community of adults with autism and severe intellectual disability

In their brief communication, Authors aim at demonstrating the association between autism onset (at a very early life stage or regressive), epilepsy , ID and worse behavioral prognosis.

 The sample is small, statistically significant results are few, however the topic can be interesting and deserves attention.

Page 2, Line 16. "Congenital Autism Specter Disorders": do Authors mean "Congenital Autism Spectrum Disorders"?

Page 4, Line 11, and Table 1. Part of the subjects who assumed anticonvulsants were not epileptic: what is the reason why they assumed this kind of drug? The same holds true for those who assumed benzodiazepines, but perhaps the use in not epileptic subjects may depend on the presence of behavioral problems. Authors should give explanation for these pharmacological treatments.

Table 1. Explanation of CASD and RASD acronyms in the figure legend would help the reader.

Please, use different symbols to indicate the statistical test that has been performed (Student's t or Fisher's exact). Indeed, the use of *and ** can be misleading, since these symbols are commonly used to indicate p≤0.05 and p≤0.01, especially when they are reported near to the significance level.

Figure 1. Lineare (Epileptic Events in CASD), Lineare (Epileptic Events in RASD): do authors mean Linear, or Linear trend?

Page 5, Line 3-6. Differences between Epilepsy No vs YES, and CASD vs RASD are never significant, there is a borderline significance only for Hyperactivity between CASD and RASD.  However, the means and medians in Epilepsy No subjects and in CASD subjects are always lower than the corresponding means and medians in Epilepsy YES and RASD subjects, respectively. Maybe (but it is not sure) that a multivariate non parametric analysis of variance, such as the  multivariate Kruskal-Wallis test, may highlight this overall difference.  Authors may try to perform such a multivariate test.

Table 2. Explanation of CASD and RASD acronyms in the figure legend would help the reader.

For the first three behavioral items,  CASD subjects seem more variable than RASD subjects. Did authors test for heteroscedasticity? May they try to attempt an explanation for this difference in variability?

Figure 2, panel B. Instead of dots and bars representing means and SD, Box-plots (possibly with dots representing the single subject values) would be much more informative.

Page 6, Discussion. May Authors try to explain why RASD is associated to epilepsy  more frequently than CASD? May be epilepsy is responsible of regression and development of autistic-like behaviors? A comment on this topic is of interest.

Round 2

Reviewer 1 Report

english must be revised

Author Response

The Autors thank you for your appreciation. 

All the listed typos have been corrected, the new sentences have been added as in the file attached 
